# Effects of an equol-containing supplement on advanced glycation end products, visceral fat and climacteric symptoms in postmenopausal women: A randomized controlled trial

**Remi Yoshikata[1,2], Khin Zay Yar Myint**📶[2]*, **Hiroaki Ohta[3], Yoko Ishigaki[4]**

**1** Hamasite Clinic, Minato-ku, Tokyo, Japan, **2** Tokyo Midtown Medical Center, Minato-ku, Tokyo, Japan, **3** Sanno Medical Center, Minato-ku, Tokyo, Japan, **4** Sendai Medical Center, Sendai-shi, Miyagi, Japan

* khinzayarmyint@gmail.com

**Data Availability Statement:** All data are available from the Harvard Dataverse (https://doi.org/10.

## Abstract

### Introduction

Equol, an isoflavone derivative whose chemical structure is similar to estrogen, is considered a potentially effective agent for relieving climacteric symptoms, for the prevention of lifestyle-related diseases, and for aging care in postmenopausal women. We investigated the effect of an equol-containing supplement on metabolism and aging and climacteric symptoms with respect to internally produced equol in postmenopausal women.

### Methods

A single-center, randomized controlled trial (registration number: UMIN000030975) on 57 postmenopausal Japanese women (mean age: 56±5.37 years) was conducted. Twenty-seven women received the equol supplement, while the remaining received control. Metabolic and aging-related biomarkers were compared before and after the 3-month intervention. Climacteric symptoms were assessed every month using a validated self-administered questionnaire in Japanese postmenopausal women.

### Results

Three months post-intervention, the treatment group showed significant improvement in climacteric symptoms compared to the control group (81% vs. 53%, respectively, p = 0.045). We did not observe any beneficial effect on metabolic and aging-related biomarkers in the intervention group. However, in certain populations, significant improvement in skin auto-fluorescence, which is a measurement of AGE skin products, and visceral fat area was observed, especially among equol producers.

### Conclusion

Women receiving equol supplementation showed improved climacteric symptoms. This study offered a new hypothesis that there may be a synergy between supplemented equol

7910/DVN/HQADC0, Harvard Dataverse, DRAFT VERSION, UNF:6:rXmd5Wltkgt0PAm3NNh+MQ== [fileUNF]).

**Funding:** The study was funded by Advanced Medical Care Co. Ltd. However, the funders had no role in study design, data collection and analysis, decision to publish, or preparation of the manuscript.

**Competing interests:** Advanced Medical Care Co. Ltd. provides administrative support to Tokyo Midtown Medical Center. This does not alter our adherence to PLOS ONE policies on sharing data and materials.

and endogenously produced equol to improve skin aging and visceral fat in certain populations.

## Introduction

Isoflavones mainly occur as glycosides of glycitein, daidzein, and genistein. In the large intestine, daidzein is converted into equol by the action of the intestinal microbiota. The production of equol from daidzein can be achieved only by a certain type of bacterial species [1]. Moreover, equol metabolism is influenced by genetic variants [2]. The number of equol producers is lower among people in Western countries than among Asians [3–6], most likely due to the influence of dietary habits. However, the younger generations in Asian countries were found to possess a lower number of equol producers, which might be due to dietary changes and the widespread use of antibiotics [7, 8].

Equol is chemically similar to estrogen. Therefore, its estrogenic actions are thought to be exerted through estrogen receptors alpha and beta [9, 10] or through G protein-coupled estrogen receptors [11, 12]. The benefits of equol producers range from relieved climacteric symptoms [13, 14] to prevention of bone density loss [15] and reduced risk of lifestyle-related diseases and cancers [16–20]. After the commercial availability of equol supplements, some of the above benefits had been tested with the use of equol supplements [21–25].

In recent years, advanced glycation end products (AGEs) have been regarded as one of the contributing factors for lifestyle-related diseases and aging. With regard to menopause, AGEs may enhance ovarian aging by increasing oxidative stress [26], initiate bone remodeling, and increase the risk of osteoporosis in postmenopausal women [27], although there has been no report on their effects on climacteric symptoms. AGEs cause stiffness and loss of elasticity through crosslinking of tissue proteins such as collagen and elastin in vessels and skin cells [28]. Therefore, the level of AGEs in the skin can be determined noninvasively by using an AGE reader by measuring skin autofluorescence or SAF [29–31].

Estradiol and isoflavones have been reported to inhibit the production of AGEs [32, 33]. As a potent isoflavone derivative and an estrogenic agent, equol is expected to exert similar effects on AGEs. However, to our knowledge, there have been no equol-based trials to study their effect on AGEs. Similarly, there have been no clinical trials on the effect of equol supplements on visceral fat. Although equol producers have been reported to be associated with reduced body and visceral fat levels [16, 34], the literature regarding the action of equol supplementation with respect to endogenously produced equol is trivial.

In this study, we aimed to investigate the effect of equol supplementation on AGEs and visceral fat areas, in addition to other metabolic and aging-related markers. We also assessed the efficacy of equol supplementation in the management of climacteric symptoms in both equol producers and nonproducers.

## Materials and methods

### Study population

A randomized controlled trial was conducted for 3 months at the Sendai Medical Center in Sendai City, Miyagi Prefecture, located in northeastern Japan. The study was conducted in compliance with the Declaration of Helsinki and approved by the Institutional Review Board of the Medical Corporation of Shinkokai. All participants provided written informed consent for participating in the study. This study was registered with the University Hospital Medical Information Network (UMIN) Clinical Trial Registry (trial registration number: UMIN000030975).

The study implementation period in the protocol was during 2017; however, due to recruitment issues, it was delayed until the beginning of 2018. The recruitment process started on September 25, 2017, using pamphlets, targeting postmenopausal women who had planned to undergo medical check-ups in January 2018. Until the scheduled date of closure on November 30, 2017, eighty-five women voluntarily participated in the study. Among them, 63 were selected based on the inclusion and exclusion criteria as follows.

Inclusion criteria: (1) Postmenopausal women (natural absence of menstruation over at least 12 months since the last menstrual period or a bilateral oophorectomy procedure performed in a woman (surgical menopause); (2) Those who could visit the clinic every month for interviews or investigations during the study period; (3) Those who could remain compliant with the daily supplementation regimen and record daily notes throughout the study period comprising 12 weeks.

Exclusion criteria: (1) Those with history of allergy to soy foods, dairy products, or Brewer's yeast; (2) Those who reported intake of medications or functional foods that could affect our study results; (3) Those administered hormone therapy or medications that could affect blood hormone levels; (4) Those considered ineligible by the investigators.

The details of the study were explained to these women, and the first 60 women to give consent to participate in the study were selected at the end of December 2017. They were listed in chronological order and assigned into two groups by simple randomization. Odd number group A received a 10 mg equol supplement and lactobionic acid daily, while even number group B received no supplement. The participants received pre-intervention investigations from January 16 to 18, 2018. The equol supplementation group started taking the supplement on the day after completion of these investigations. All the participants had a regular follow-up every month for 3 months until April 20, 2018. In the equol supplementation group, one woman dropped out, and two women reported that they had occasionally used the equol supplement before the intervention. Consequently, 57 women (48 to 69 years of age) were eventually included for analysis in the study (Fig 1). The nurses at the Sendai Medical Center generated the random allocation sequence, enrolled participants, and assigned participants to interventions. The registered dietitian was responsible for explanation, distribution and monitoring supplements to the intervention group.

## Study treatments

Twenty-seven women belonging to the equol supplementation group received 10 mg of oral equol-containing supplement per day, composed of 98% S-equol, 2% daidzein, 0.2% glycitein, and 0.1% genistein extracted from fermented soybeans (product name: FlavoCel EQ-5, Daicel Corporation, Tokyo, Japan).

## Determination of equol-producer status

Prior to the intervention, early morning urine samples were collected from all participants. Urinary equol was measured using an immunochromatographic strip (Soy Check, Healthcare Systems Co., Ltd), as described in a previous study [15]. Individuals were considered equol producers if their urinary equol level was higher than 1.0 μM, as described in previous studies [3, 4].

## Primary outcome measures

Body height and weight were measured using a height weight scale (A & D Company Limited, Tokyo, Japan). Overnight fasting blood samples were obtained to determine the levels of triglycerides (TG), total cholesterol (TC), low-density lipoprotein (LDL) cholesterol, high-density

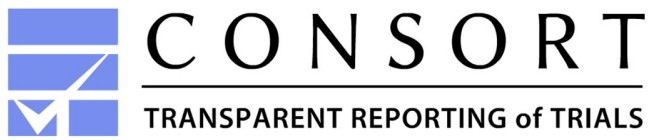

**Enrollment**

Assessed for eligibility (n=85)

Excluded (n=25)
- ◆  Not meeting inclusion criteria (n=22)
- ◆  Declined to participate (n=0)
- ◆  Other reasons (n=3)

Enrolled (n=60)

**Allocation**

Allocated to intervention (n=30)
- ◆ Received allocated intervention (n=30)
- ◆ Did not receive allocated intervention (n=0)

Allocated to intervention (n=30)
- ◆ Received allocated intervention (n=30)
- ◆ Did not receive allocated intervention (n=0)

**Follow-Up**

Lost to follow-up (no apparent reason) (n=1)

Discontinued intervention (n=0)

Lost to follow-up (n=0)

Discontinued intervention (n=0)

**Analysis**

Analysed   (n=27   )
- ◆ Excluded from analysis (history of equol use before participation) (n=2)

Analysed (n=30)
- ◆ Excluded from analysis    (n=0)

**Fig 1. CONSORT flow diagram.**

lipoprotein (HDL) cholesterol, hemoglobin A1c (HbA1c), uric acid (UA), intact parathyroid hormone (PTH), and 25-hydroxy vitamin D. To assess the degree of arterial stiffness (arterio-sclerosis), brachial-ankle pulse wave velocity (baPWV) was measured using vascular ultra-sound (Fukuda Denshi, Tokyo, Japan). Visceral fat area was measured using computed tomography (CT). The level of AGEs was determined by measuring the skin autofluorescence on the volar side of the forearm using an AGE reader (DiagnOptics, Groningen, Netherlands) as described previously [28]. Autofluorescence was defined as the average fluorescence per nm over the entire emission spectrum (420–600 nm) as the ratio of the average fluorescence per nm over the 300-420-nm range [28]. Age-adjusted SAF levels (z scores) were calculated for each woman based on the total population. The above measurement of parameters was performed at baseline and 3 months postintervention.

## Secondary outcome measures

Self-administered questionnaires were used for the assessment of climacteric symptoms using the Climacteric Scale developed by the Japan Society of Obstetrics and Gynecology (Table 1). The questionnaire contained 21 items scored on a 4-point scale (Never = 0, mild = 1, moderate = 2, severe = 3). The total score represented the overall severity of the symptoms. These questionnaires were administered during the monthly follow-ups for 3 months. Starting from the first month of the intervention, the overall improvement in symptoms was also assessed using the following four responses: 'a lot', 'somewhat', 'no change', and 'worse'. This technique was validated and widely used in the Japanese population.

## Treatment adherence and monitoring of adverse effects

During the monthly follow-ups, the staff interviewed adherence to equol supplements, such as frequency and dose, as well as clinical signs and symptoms of adverse effects, to ensure compliance and to make timely decisions on cessation of the supplement. No participant reported any adverse effects during the course of the study.

## Statistical analyses

All statistical analyses were performed using IBM SPSS 19 statistical software (IBM Japan, Minato-ku, Tokyo, Japan). The Mann-Whitney test was used to compare the differences in continuous data, and the chi-square test was used to compare categorical data between the equol supplementation group and the control group. For the assessment of quantitative changes in the metabolic and aging biomarkers before and after intervention, we used the Wilcoxon sign rank test. To compare the proportions of people with improved metabolic and aging biomarkers between the control and equol supplementation groups, we used the chi-square test and Fisher's exact test. Additionally, we assessed the change in results with respect to equol exposure by categorizing the groups into four categories: 1) equol producers consuming equol supplements, 2) equol nonproducers consuming equol supplements, 3) quol producers without equol supplements, and 4) equol nonproducers without supplements. The extended Mantel-Haenszel chi square for linear trend was used to examine the association between postintervention improvement (dependent variable) and equol exposure (independent variable).

Changes in the severity (total score) of climacteric symptoms over time, i.e., at baseline and 1 month, 2 months and 3 months post equol intervention, were analyzed using a two-way repeated measure analysis of variance (ANOVA). Analysis of the studentized residuals showed that there was normality, as assessed by the Shapiro-Wilk test of normality and no outliers, as assessed by no studentized residuals greater than ± 3 standard deviations. There was sphericity

**Table 1. Climacteric symptom rating scale by the Japan society of obstetrics and gynecology.**

| | Item | Score |
|---|---|---|
| Q1 | 1. Facial skin blushing and upper body (hot flashes) | Never = 0, mild = 1, moderate = 2, severe = 3 |
| Q2 | 2. Easy to sweat (sweating) | Never = 0, mild = 1, moderate = 2, severe = 3 |
| Q3 | 3. Difficulty getting to sleep (insomnia) | Never = 0, mild = 1, moderate = 2, severe = 3 |
| Q4 | 4. Difficulty staying asleep (light sleep) | Never = 0, mild = 1, moderate = 2, severe = 3 |
| Q5 | 5. Irritability | Never = 0, mild = 1, moderate = 2, severe = 3 |
| Q6 | 6. Anxiety | Never = 0, mild = 1, moderate = 2, severe = 3 |
| Q7 | 7. Often irritated by trifles (anxious trifles) | Never = 0, mild = 1, moderate = 2, severe = 3 |
| Q8 | 8. Feeling unhappy or depressed (depressive mood), | Never = 0, mild = 1, moderate = 2, severe = 3 |
| Q9 | 9. Fatigue | Never = 0, mild = 1, moderate = 2, severe = 3 |
| Q10 | 10. Eye strain | Never = 0, mild = 1, moderate = 2, severe = 3 |
| Q11 | 11. Memory problems (forgetfulness) | Never = 0, mild = 1, moderate = 2, severe = 3 |
| Q12 | 12. Dizziness | Never = 0, mild = 1, moderate = 2, severe = 3 |
| Q13 | 13. Palpitations | Never = 0, mild = 1, moderate = 2, severe = 3 |
| Q14 | 14. Chest tightness | Never = 0, mild = 1, moderate = 2, severe = 3 |
| Q15 | 15. Headache | Never = 0, mild = 1, moderate = 2, severe = 3 |
| Q16 | 16. Neck stiffness | Never = 0, mild = 1, moderate = 2, severe = 3 |
| Q17 | 17. Backache and low back pain | Never = 0, mild = 1, moderate = 2, severe = 3 |
| Q18 | 18. Joint pain | Never = 0, mild = 1, moderate = 2, severe = 3 |
| Q19 | 19. Cold hands and feet | Never = 0, mild = 1, moderate = 2, severe = 3 |
| Q20 | 20. Numbness in the legs or arms | Never = 0, mild = 1, moderate = 2, severe = 3 |
| Q21 | 21. Sensitive to sounds | Never = 0, mild = 1, moderate = 2, severe = 3 |
| Q22 | Overall improvement in symptoms (not included in the baseline questionnaire) | Worse = -1, No change = 0, Somewhat = 1, A lot = 2 |

for the interaction term, as assessed by Mauchly's test of sphericity ($p > .05$). Additionally, post hoc pairwise comparisons between equol intervention and control groups as well as between equol producers and nonproducers were conducted using the Bonferroni correction. Improvements in climacteric symptoms in the control and equol supplementation groups were compared using the chi-square test and Fisher's exact test for proportions. All tests were two-sided, and the statistical significance was set to $p<0.05$.

# Results

## Characteristics of participants

The baseline characteristics of the equol supplementation group and the control group showed no statistically significant difference, except for drinking habits and supplement use (Table 2). Among the 57 women analyzed, 13 were equol producers (22.8%).

## Changes after 3 months

There were no missing data in primary outcome measures among the 57 women analyzed. As shown in Table 3, both groups exhibited a decrease in the levels of HDL cholesterol and 25-OH vitamin D and an increase in the level of PTH. In the equol supplementation group, a significant reduction was observed in the levels of LDL and total cholesterol. In the control group, the visceral fat level was significantly reduced after 3 months. Although such changes were observed at baseline and after 3 months in both groups, there was no significant difference between the two groups with respect to these changes. Next, the proportion of women in the two groups displaying changes from baseline was compared (Table 4). After 3 months, more women in the control group were found to have worse PWV values than those in the equol intervention group (26.7% vs. 3.7%, x-squared (2, 27) = 6.33, 95% CI [0.056, 0.403], p = 0.042).

## Changes with regards to equol exposure categories on ad hoc analysis

Initially, no significant difference was observed among the four equol exposure categories: 1) both intrinsic and extrinsic equol exposure, 2) exclusively extrinsic equol exposure, 3) exclusively intrinsic equol exposure, and 4) no equol exposure. However, in the subgroup analysis,

**Table 2. Basic characteristics of the equol supplementation group and control group.**

| Basic characteristics | All participants (n = 57) | Equol intervention (n = 27) | Control group (n = 30) | p-value |
|---|---|---|---|---|
| Age | 56 (48–69) | 56 (48–69) | 56 (49–69) | 0.697[a] |
| Equol producers | 13 (22.8%) | 4 (14.8%) | 9 (30%) | 0.172 [a] |
| Body-mass index | 21.4 (16.5–27.9) | 21.2 (16.5–26.9) | 21.5 (18.2–27.9) | 0.455[b] |
| Smoking habit | | | | 0.617 [b] |
| No | 54 (94.74%) | 26 (96.30%) | 28 (93.33%) | |
| Past or current smoker | 3 (5.26%) | 1 (3.70%) | 2 (6.67%) | |
| Drinking habit | | | | **0.009** [b] |
| No | 20 (10.53%) | 15 (3.70%) | 5 (16.67%) | |
| Sometimes | 23 (40.35%) | 7 (25.93%) | 16 (53.33%) | |
| Everyday | 14 (24.56%) | 5 (18.52%) | 9 (30.00%) | |
| Exercise habit (at least 2 times per week) | 17 (30%) | 8 (29.6%) | 9 (30%) | 0.764 [b] |
| Current medication use | | | | |
| Anti-hypertensive | 8 (14.04%) | 5 (18.52%) | 3 (10.00%) | 1.000 [b] |
| Anti-diabetes | 1 (1.75%) | 0 (0.00%) | 1 (3.33%) | 1.000 [b] |
| Cholesterol lowering drugs | 4 (7.02%) | 3 (11.11%) | 1 (3.33%) | 0.214 [b] |

Continuous values are shown as medians (ranges) and categorical values are shown as numbers and proportions (%).

[a]: p-value for Mann-Whitney's tests.

[b]: p-value for chi-squared tests.

Statistically significant differences (p-value <0.05) are shown underlined in bold.

**Table 3. Comparison of quantitative changes in metabolic and aging biomarkers between the equol supplementation group and the control group.**

| Primary measures | (n) | Baseline Median (IQR) | 12 weeks Median (IQR) | p-value[a] | Median change | p-value[b] |
|---|---|---|---|---|---|---|
| **Skin autofluorescence** | | | | | | 0.13 |
| Control | (30) | **2.2** (2.0–2.3) | **2.3** (2.1–2.4) | 0.268 | 0.05 | |
| Equol | (27) | **2.1** (1.9–2.4) | **2.2** (1.9–2.3) | 0.402 | 0 | |
| **Visceral fat** | | | | | | 0.234 |
| Control | (30) | **70.1** (32.5–98.0) | **63.8** (28.5–92.1) | **0.023** | -5.35 | |
| Equol | (27) | **46.5** (33.8–85.7) | **49.5** (28.5–83.2) | 0.136 | -2.5 | |
| **PWV** | | | | | | |
| Control | (30) | **1251** (1118–1398) | **1228** (1167–1422) | 0.877 | 7 | |
| Equol | (27) | **1224** (1171–1367) | **1294** (1193–1431) | 0.746 | -5.5 | |
| **SBP** | | | | | | 0.227 |
| Control | (30) | **118** (101–125) | **114** (99–128) | 0.877 | 1.5 | |
| Equol | (27) | **114** 104–126) | **108** (97–118) | 0.206 | -1 | |
| **DBP** | | | | | | 0.198 |
| Control | (30) | **70** (63–82) | **72** (69–84) | **0.016** | 4 | |
| Equol | (27) | **70** (66–74) | **70** (62–77) | 0.979 | 0 | |
| **Total cholesterol** | | | | | | 0.701 |
| Control | (30) | **216** (196–240) | **208** (199–226) | 0.065 | 0 | |
| Equol | (27) | **214** (200–266) | **208** (195–256) | **0.031** | -9 | |
| **Triglycerides** | | | | | | 0.209 |
| Control | (30) | **74** (52–111) | **71** (50–102) | 0.779 | 0.5 | |
| Equol | (27) | **70** (48–93) | **84** (58–106) | **0.038** | 9 | |
| **HDL-C** | | | | | | 0.917 |
| Control | (30) | **71** (62–93) | **67** (59–89) | **0.005** | -3 | |
| Equol | (27) | **77** (65–83) | **71** (62–79) | **0.001** | -4 | |
| **LDL-C** | | | | | | 0.362 |
| Control | (30) | **121** (94–143) | **117** (100–144) | 0.863 | 0.5 | |
| Equol | (27) | **133** (101–162) | **134** (102–160) | 0.226 | -1 | |
| **Uric acid** | | | | | | 0.245 |
| Control | (30) | **4.6** (4.1–5.9) | **4.3** (3.8–5.5) | 0.088 | -0.2 | |
| Equol | (27) | **4.1** (3.9–4.9) | **4.4** (3.9–5.0) | 0.989 | 0 | |
| **HbA1c** | | | | | | 0.354 |
| Control | (30) | **5.7** (5.5–5.8) | **5.6** (5.5–5.8) | 0.679 | -0.05 | |
| Equol | (27) | **5.8** (5.6–6.0) | **5.8** (5.6–6.0) | 0.514 | 0 | |
| **PTH** | | | | | | 0.074 |
| Control | (30) | **45.5** (36.0–56.8) | **54.5** (45.8–71.0) | **p<0.001** | 10.5 | |
| Equol | (27) | **45.0** (32.0–51.0) | **56.0** (47.0–65.0) | **p<0.002** | 14 | |
| **Vitamin D** | | | | | | 0.342 |
| Control | (30) | **13.7** (12.9–17.6) | **12.0** (8.8–17.6) | 0.071 | -2.45 | |
| Equol | (27) | **15.9** (13.5–17.8) | **12.4** (10.7–13.9) | **0.002** | -3.6 | |
| **BMI** | | | | | | 0.655 |
| Control | (30) | **21.6** (20.0–25.1) | **21.5** (20.1–24.5) | 0.066 | -0.119 | |
| Equol | (27) | **21.2** (20.0–23.7) | **21.2** (20.0–23.2) | 0.186 | -0.0946 | |

IQR Interquartile range.

p-value[a] Wilcoxon sign rank test.

p-value[b] Mann-Whitney test.

**Table 4. Comparison of changes between the equol supplementation group and the control group.**

| Primary measures | Improved | | Changes within normal limits | | Worsen | | p-value |
|---|---|---|---|---|---|---|---|
| | n | % | n | % | n | % | |
| **Skin autofluorescence** | | | | | | | 0.589 |
| Control | 1 | 3.33% | 27 | 90.00% | 0 | 0.00% | |
| Equol | 1 | 3.70% | 25 | 92.59% | 1 | 3.70% | |
| **Visceral fat** | | | | | | | 0.165 |
| Control | 4 | 13.33% | 23 | 76.67% | 3 | 10.00% | |
| Equol | 2 | 7.41% | 25 | 92.59% | 0 | 0.00% | |
| **PWV** | | | | | | | **0.042** |
| Control | 3 | 10.00% | 19 | 63.33% | 8 | 26.67% | |
| Equol | 6 | 22.22% | 20 | 74.07% | 1 | 3.70% | |
| **SBP** | | | | | | | 0.584 |
| Control | 3 | 10.00% | 24 | 80.00% | 3 | 10.00% | |
| Equol | 4 | 14.81% | 22 | 81.48% | 1 | 3.70% | |
| **DBP** | | | | | | | 0.393 |
| Control | 1 | 3.33% | 27 | 90.00% | 2 | 6.67% | |
| Equol | 1 | 3.70% | 26 | 96.30% | 0 | 0.00% | |
| **LDL** | | | | | | | 0.658 |
| Control | 9 | 30.00% | 13 | 43.33% | 8 | 26.67% | |
| Equol | 11 | 40.74% | 9 | 33.33% | 7 | 25.93% | |
| **HDL** | | | | | | | |
| Control | 0 | 0.00% | 30 | 100.00% | 0 | 0.00% | |
| Equol | 0 | 0.00% | 30 | 111.11% | 0 | 0.00% | |
| **Triglycerides** | | | | | | | 0.469 |
| Control | 1 | 3.33% | 25 | 83.33% | 4 | 13.33% | |
| Equol | 0 | 0.00% | 25 | 92.59% | 2 | 7.41% | |
| **Vitamin D** | | | | | | | 0.776 |
| Control | 9 | 30.00% | 0 | 0.00% | 21 | 70.00% | |
| Equol | 7 | 25.93% | 0 | 0.00% | 20 | 74.07% | |
| **PTH** | | | | | | | 0.779 |
| Control | 0 | 0.00% | 19 | 63.33% | 11 | 36.67% | |
| Equol | 0 | 0.00% | 19 | 70.37% | 8 | 29.63% | |
| **Uric acid** | | | | | | | |
| Control | 0 | 0.00% | 30 | 100.00% | 0 | 0.00% | |
| Equol | 0 | 0.00% | 27 | 100.00% | 0 | 0.00% | |
| **HbA1c** | | | | | | | |
| Control | 0 | 0.00% | 30 | 100.00% | 0 | 0.00% | |
| Equol | 0 | 0.00% | 27 | 100.00% | 0 | 0.00% | |

p-value Chi-squared test or Fisher exact test.

the extended Mantel-Haenszel chi square for linear trend showed significant linear trends of improvement in visceral fat area and skin autofluorescence upon exposure to equol.

Fig 2A shows the proportion of women with improved visceral fat area after 3 months, excluding those women on regular exercise and those consuming lipid-lowering medications (n = 50). Women with a habit of exercising regularly were excluded from the analysis because regular exercise is the most effective strategy for reducing visceral fat [35]. Visceral fat areas were found to be increased in 6 out of 14 people with no equol exposure (42.9%), 1 out of 7

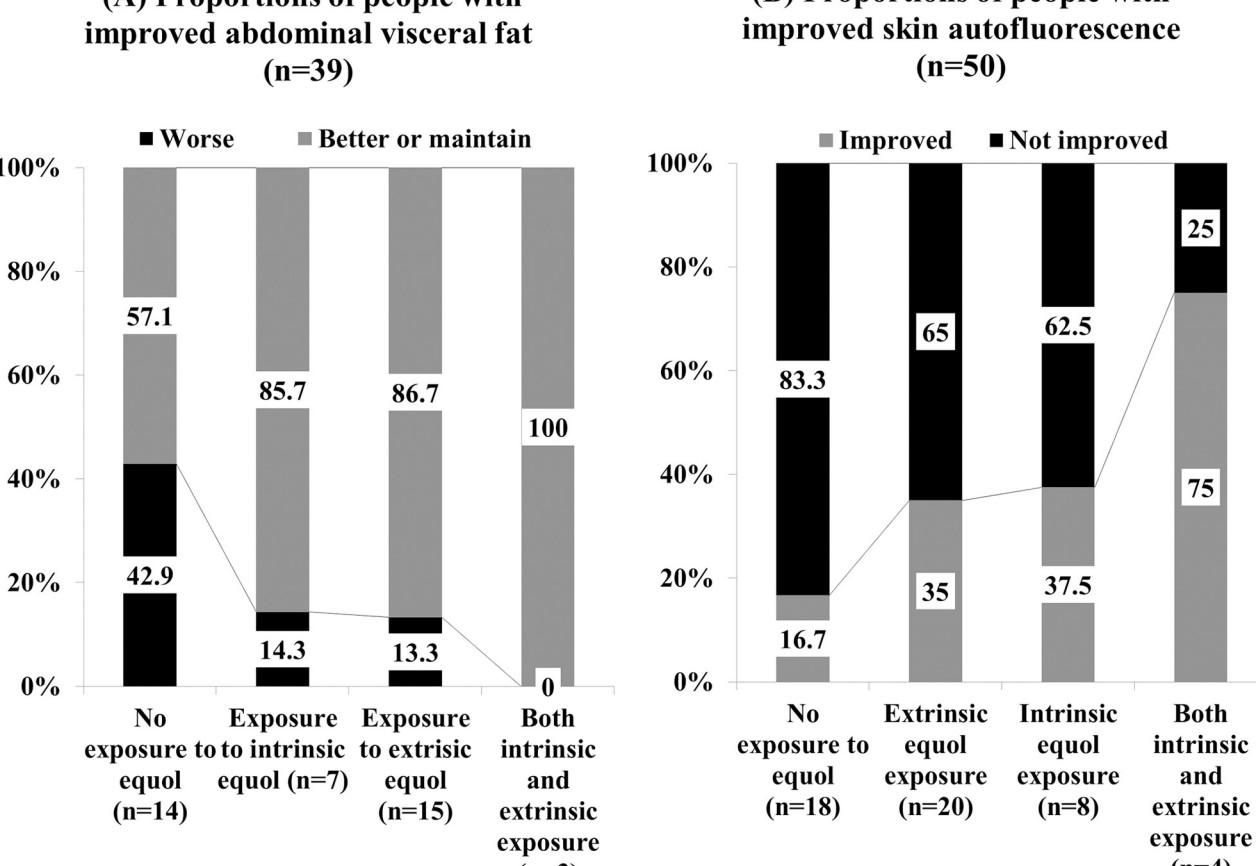

**Fig 2. Effects of intrinsic and extrinsic equol exposure after 3 months.** A synergy between supplemented equol and endogenously produced equol on skin autofluorescence and visceral fat areas.

people in the exclusively intrinsic exposure group (14.3%), 2 out of 15 people in the exclusively extrinsic exposure group (13.3%), and nobody in the intrinsic and extrinsic exposure group (0%). The extended Mantel-Haenszel chi square for the linear trend showed significant linear trends (p = 0.023) of improvement in visceral fat area in response to equol exposure.

Fig 2B shows the proportion of people among 39 women with improved skin autofluorescence after 3 months, measured using an AGE reader, excluding women with a current history of medications such as lipid-lowering agents and antidiabetic agents. Women consuming the aforementioned drugs were excluded because these medications could affect the results of skin autofluorescence [36, 37]. Skin autofluorescence was found to be improved in 3 out of 18 people with no equol exposure (16.7%), in 7 out of 20 people in the exclusively extrinsic exposure group (35%), in 3 out of 8 people in the exclusively intrinsic exposure group (37.5%), and in 3 out of 4 people in the group with both intrinsic and extrinsic exposure (75%). The extended Mantel-Haenszel chi square for the linear trend showed significant linear trends (p = 0.044) in the improvement of skin autofluorescence in response to equol supplementation.

### Changes in climacteric symptom scores

Two-way repeated measures ANOVA indicated a significant time effect ($F_{(3,159)}$ = 4.055, p = 0.009) and group x time interaction ($F_{(3,159)}$ = 3.531, p = 0.017), suggesting that the

climacteric symptoms improved during the study period in the equol supplementation group and that the group effect varied with time. A post hoc pairwise comparison using the Bonferroni correction showed a decrease in total climacteric symptom scores between the initial assessment and follow-up assessment at 2 months after intervention (16.44 vs 12.22, respectively), which was statistically significant (p = 0.014). Furthermore, the decrease in total climacteric symptom scores did reach significance when comparing the initial assessment to a follow-up assessment taken 3 months after the intervention (16.44 vs 11.33, p = .002). Therefore, we can conclude that the ANOVA results indicated that the difference in climacteric symptom improvement in the equol supplementation group was more pronounced with time. However, there was no interaction between the groups with respect to time and equol producer status, indicating that equol supplementation might be beneficial regardless of the equol producer status (Fig 3A).

### Changes in the self-reported overall improvement of symptom scores

The proportion of women who reported overall improvement of symptoms in the equol supplement group was 56%, compared to 40% in the control group 1 month after the intervention. A marginal increase in this proportion was observed in the control group at 2 months postintervention (53%), but no change was seen at 3 months. However, these proportions significantly increased in the equol supplementation group, especially at 3 months postintervention, where 81% of the women experienced improvement in climacteric symptoms (Fig 3B).

## Discussion

In this study, only ~22.8% of the population was equol producers, which is lower than that previously reported [6, 15]. The gut microbiota is established immediately after birth [38]. Consequently, it is shaped according to individual dietary and lifestyle habits. Our previous study showed that equol production was associated with the type of diet as well as with lifestyle habits [39]. Therefore, individual differences in the composition of gut microbiota might be responsible for different results reported in different studies. In recent years, microorganisms that can produce equol have been identified more. Using genetic engineering techniques, these microorganisms could be used for the formulation of functional foods.

A significant decrease in LDL levels was observed in the equol supplementation group, which in turn might be responsible for the lower total cholesterol level. Furthermore, a reduction in vitamin D and HDL cholesterol and increased PTH levels were noted in both groups. The reason for the reduction in the vitamin D level might be due to the reduced production of vitamin D, as exposure to ultraviolent light was reduced during the winter in northeastern Japan when this study was conducted. As vitamin D levels are inversely associated with PTH levels, PTH levels might be increased as a consequence. In some studies, the HDL level was found to be positively associated with vitamin D levels in postmenopausal women [40, 41]. Therefore, the decrease in vitamin D levels might be responsible for the decrease in HDL cholesterol levels in both groups. Although the exact mechanism for this association has not been fully elucidated, further studies are warranted to explore the role of vitamin D in preventing cardiovascular diseases.

The above findings are contradictory to the results obtained in previous studies of postmenopausal women administered equol supplementation for one year [23, 25]. One study found that equol supplementation helped to reduce bone mineral density [23]. Another study found improvements in cardiovascular- and bone density-related biomarkers such as arterial stiffness, triglycerides, urinary NTX, and intact PTH levels [25]. The discrepancy in the results might be because the current study was conducted as a short-term intervention among

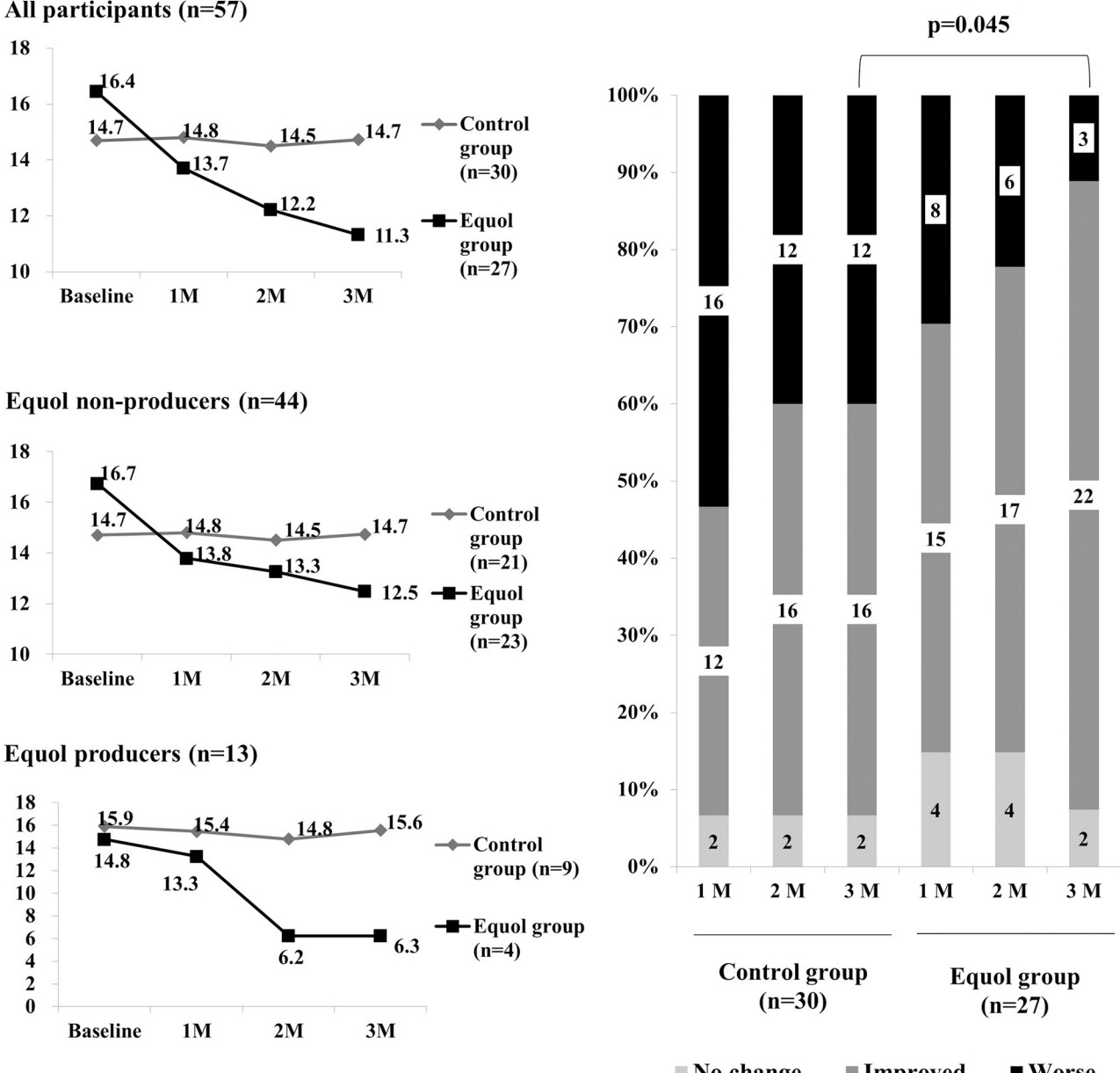

**Fig 3. Effect of equol on climacteric symptom scores.** Equol supplementation could be beneficial to both equol producers and nonproducers for relieving climacteric symptoms.

relatively healthy individuals, while previous studies were long-term and were conducted among high-risk individuals with pre-existing morbidities. Concordantly, in the current study, we found that arterial stiffness—measured in terms of baPWV level—was improved in women with high baseline baPWV values.

With respect to visceral fat area, there have been no randomized controlled trials on the effects of equol supplementation. However, one study reported an association between

reduced visceral fat levels and equol producer status [16]. The potential mechanism of action of equol might be due to its estrogenic property. It acts by modulating body fat deposition [42] or by directly acting on the genes involved in energy metabolism [34]. In this study, it was found that greater exposure to equol was less likely to be associated with unhealthy visceral fat levels in postmenopausal women without regular exercise habits at 3 months postintervention. At least one kind of exposure to equol, either intrinsically via equol production or extrinsically via equol supplementation, might aid in preventing the worsening of visceral fat levels in post-menopausal women. Possessing both exposures, that is, an equol producer consuming an equol supplement, could be the most effective mechanism for maintaining visceral fat levels. Consequently, it is probable that equol could help reduce central obesity-related diseases in postmenopausal women who could not exercise regularly due to physical morbidities.

In recent years, AGEs have been considered to play an important role in aging. The accumulation of AGEs in the skin can be determined by measuring skin autofluorescence or SAF, and values are age-adjusted [29–31]. As AGE production requires estrogen, the estrogen level postmenopause might affect AGEs [32, 43]. The effects of isoflavones on AGEs, especially the inhibition of AGE formation, have been reported [33]. Isoflavones act via 3 mechanisms: 1) prevention of the Maillard reaction, *i.e.*, prevention of sugar-protein compound formation; 2) inhibition of the oxidation of sugar, fat and amino acids, thereby preventing the formation of carbonyl groups; and 3) prevention of the conversion of Schiff bases into carbonyl groups. As equol is a metabolite of isoflavones, it might exert similar effects as those of isoflavones. Moreover, as equol exerts anti-aging and anti-arteriosclerotic effects on the skin [44], the skin auto-fluorescence values could also be affected.

With respect to the improvement in climacteric symptoms, both equol producers and non-producers showed similar levels of improvement in the treatment group. Previous studies of equol efficacy on climacteric symptoms have focused on equol nonproducers. In this study, we found that equol supplementation further improved climacteric symptoms in equol producers.

Despite the many positive findings, our study had some limitations. First, it was a small-scale, single-center study, which limited the generalizability. Second, we randomly assigned the intervention and control groups without considering the equol producer status. Although there was no statistically significant difference between the two groups with respect to their equol producer status, it would have been better if we had assigned equal proportions of equol producers and nonproducers to both groups. Third, although we assessed food habits at base-line, we could not assess the changes in food habits after 3 months. Therefore, the type of diet and the amount of isoflavone intake might affect the outcomes [45, 46]. However, the effect of diet is considered trivial, as diet is not the sole determinant of equol production and metabolism [1, 2, 47]. Furthermore, it is not clear whether the nonproducer status can be converted to producer status without isoflavone intervention within three months, as previous studies have shown that the equol production phenotype is relatively stable without dietary intervention [48–50].

Another important limitation of this study was that we were unable to calculate the sample size statistically since we could not find any previous clinical trial on the effects of equol or iso-flavone on AGEs or visceral fat to use as a reference for sample size calculation. Therefore, we applied the general rule of thumb for sample size determination for this study, which included 30 postmenopausal women in each group. A post hoc power analysis was conducted using R statistical software 4.1. The analysis revealed that the statistical power of the t-test was 95% for a moderate effect size of 0.5 for the sample size of 27 in each group, which was significant at the 5% level (two tailed). In addition, we found that there may be a dose-response relationship depending on individual equol production ability only after analyzing the data. We were also

unaware that exercise or certain medications might have a significant effect on AGEs or visceral fat before the study due to insufficient literature review. These factors should have been considered in calculating the sample size before starting the study.

Despite the aforementioned limitations, this is the first study to report the effect of equol supplementation on visceral fat area and AGEs. It also provided new insight that the action of equol possesses a dose-response relationship, depending on the inherent equol production ability of the population. Greater equol exposure might be more beneficial in correcting visceral adiposity in people without regular exercise and in protecting against aging. Moreover, since the equol producing ability of an individual was found to be associated with microbial diversity [39], the gut environment might favor the absorption of the equol supplement.

Furthermore, we found that equol supplementation could be beneficial to both equol producers and nonproducers for relieving climacteric symptoms. Equol producers tend to have lower severity of climacteric symptoms, with improvement in symptoms associated with the consumption of an equol supplement. Most of the previous studies focused on the benefits of equol in non-equol producer women. Similar to our previous prospective study on postmenopausal women using equol supplementation for a year [25], this study also indicated the beneficial effect of equol supplementation on the relief of climacteric symptoms in equol producer women.

## Conclusion

From this study, we found that equol supplementation had the potential to improve visceral fat area and advanced glycation end product production in certain postmenopausal women. The benefits of equol supplementation might be enhanced in women with inherent equol producing ability. Additionally, equol supplementation led to an overall improvement in climacteric symptoms, regardless of the equol producer status. Consequently, equol might be a potential alternative to HRT for relieving climacteric symptoms. Moreover, it can also be used for the prevention of lifestyle-related diseases and for aging care in postmenopausal women. However, caution should be applied since these preliminary findings warrant further investigation to confirm the obtained results.

## Supporting information

**S1 File. Study protocol in Japanese.**
(PDF)

**S2 File. Study protocol in English translation.**
(PDF)

**S3 File. CONSORT 2010 checklist_equol research.**
(PDF)

**S1 Data. Figs 1, 2 and 3.**
(ZIP)

## Acknowledgments

We would like to acknowledge all the women who willingly came forward to participate in the study.

## Author Contributions

**Conceptualization:** Remi Yoshikata, Yoko Ishigaki.

**Data curation:** Khin Zay Yar Myint.

**Formal analysis:** Khin Zay Yar Myint.

**Funding acquisition:** Remi Yoshikata.

**Investigation:** Remi Yoshikata, Yoko Ishigaki.

**Methodology:** Khin Zay Yar Myint.

**Project administration:** Remi Yoshikata.

**Supervision:** Hiroaki Ohta.

**Visualization:** Khin Zay Yar Myint.

**Writing – original draft:** Remi Yoshikata.

**Writing – review & editing:** Khin Zay Yar Myint.

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
