## [Decision Letter · Decision Letter 0]

25 May 2021

PONE-D-21-06650

Equol effects on AGE skin products, visceral fat, and climacteric symptoms in post-menopausal women

PLOS ONE

Dear Dr. Myint,

Thank you for submitting your manuscript to PLOS ONE. After careful consideration, we feel that it has merit but does not fully meet PLOS ONE’s publication criteria as it currently stands. Therefore, we invite you to submit a revised version of the manuscript that addresses the points raised during the review process.

We look forward to receiving your revised manuscript.

Kind regards,

Walid Kamal Abdelbasset, Ph.D.

Academic Editor

PLOS ONE

Journal Requirements:

2.We note that you have indicated that data from this study are available upon request. PLOS only allows data to be available upon request if there are legal or ethical restrictions on sharing data publicly. For information on unacceptable data access restrictions, please see http://journals.plos.org/plosone/s/data-availability#loc-unacceptable-data-access-restrictions.

6.Thank you for stating the following financial disclosure:

 "Advanced Medical Care Co. Ltd."       

7.Thank you for stating the following in the Competing Interests section:

"Advanced Medical Care Co. Ltd. provides administrative support to Tokyo Midtown Medical Center."

Reviewers' comments:

Reviewer's Responses to Questions

**Comments to the Author**

1. Is the manuscript technically sound, and do the data support the conclusions?

Reviewer #1: Partly

Reviewer #2: Partly

2. Has the statistical analysis been performed appropriately and rigorously? 

Reviewer #1: Yes

Reviewer #2: Yes

3. Have the authors made all data underlying the findings in their manuscript fully available?

Reviewer #1: No

Reviewer #2: Yes

4. Is the manuscript presented in an intelligible fashion and written in standard English?

Reviewer #1: Yes

Reviewer #2: No

5. Review Comments to the Author

Reviewer #1: A two-arm randomized controlled study (n=57) was conducted to compare pre and post intervention of Equol on climacteric symptoms in post-menopausal women. At 3-month follow-up the intervention group showed statistically significant improvements in climacteric symptoms compared to placebo.

Minor revisions:

1- Table 2: Typographical error in footnote: Replace A with a

2- Line 212: Provide 95% confidence intervals for the proportions 26.7% and 3.7%.

3- Table 3: Typographical error in footnote: Mann-Whitney test.

4- Indicate if any adverse events occurred during the course of the study.

5- State and justify the study’s target sample size with a pre-study statistical power calculation. The power calculation should include: sample size, alpha level (indicating one or two-sided), minimal detectable difference and statistical testing method.

Reviewer #2: Paper titled (Equol effects on AGE skin products, visceral fat, and climacteric symptoms in postmenopausal women) by Yoshikata et al. demonstrated the effect of Equol, an isoflavone derivative with similar chemical structure like estrogen.

Although this is an interesting topic with positive outcome. However, the style of writing is hard to be comprehended or followed.

This was starting from the Title until the end of Ms

For example: title: Equol effect on.... : this is better saif The effect of Equol, an isoflavone, on .....

This is since this flavone is not very common or known.

Kindly consult a colleague proficient in English and academic writing to help you better demonstrate your idea

6. PLOS authors have the option to publish the peer review history of their article (what does this mean?). If published, this will include your full peer review and any attached files.

Reviewer #1: No

Reviewer #2: **Yes: **Sawsan Zaitone

---

## [Author Response · Author response to Decision Letter 0]

8 Jun 2021

 �We have revised accordingly.

2.We note that you have indicated that data from this study are available upon request. PLOS only allows data to be available upon request if there are legal or ethical restrictions on sharing data publicly. For information on unacceptable data access restrictions, please see http://journals.plos.org/plosone/s/data-availability#loc-unacceptable-data-access-restrictions.

 �We have uploaded the minimal anonymized data set in Harvard Dataverse and provided the link. 

 �We have revised accordingly.

We have moved it from the declaration section to the method section.

 �We have moved it from the declaration section to the method section (line 88-93).

6.Thank you for stating the following financial disclosure:

 "Advanced Medical Care Co. Ltd." 

We have included this amended Role of Funder statement in our cover letter.

7. Thank you for stating the following in the Competing Interests section:

"Advanced Medical Care Co. Ltd. provides administrative support to Tokyo Midtown Medical Center."

We have included this Competing Interests statement in our cover letter.

 

Comments to the Author

5. Review Comments to the Author

 Reviewer #1: A two-arm randomized controlled study (n=57) was conducted to compare pre and post intervention of Equol on climacteric symptoms in post-menopausal women. At 3-month follow-up the intervention group showed statistically significant improvements in climacteric symptoms compared to placebo.

Minor revisions:

1- Table 2: Typographical error in footnote: Replace A with a

We have revised accordingly.

2- Line 212: Provide 95% confidence intervals for the proportions 26.7% and 3.7%.

We have provided the 95% confidence interval for the proportions 26.7% and 3.7% in line 221 as follows.

Post 3 months, more women in the control group were found to have worsen PWV values compared to those in the equol intervention group (26.7% vs. 3.7%, x-squared (2, 27) = 6.33, 95% CI [0.056, 0.403], p=0.042).

3- Table 3: Typographical error in footnote: Mann-Whitney test.

We have revised accordingly.

4- Indicate if any adverse events occurred during the course of the study.

We added information on the adverse events during the course of the study in line 172 as follows.

No participant reported any adverse effects during the course of the study.

5- State and justify the study’s target sample size with a pre-study statistical power calculation. The power calculation should include: sample size, alpha level (indicating one or two-sided), minimal detectable difference and statistical testing method.

We have state and justified the study’s target sample size with a pre-study statistical power calculation in line 371-373 as follows.

A post hoc power analysis was conducted using R statistical software 4.1. The analysis revealed that the statistical power of t- test was 95% for a moderate effect size of 0.5 for the sample size of 27 in each group, as significant at the 5% level (two tailed). 

Reviewer #2: Paper titled (Equol effects on AGE skin products, visceral fat, and climacteric symptoms in postmenopausal women) by Yoshikata et al. demonstrated the effect of Equol, an isoflavone derivative with similar chemical structure like estrogen.

Although this is an interesting topic with positive outcome. However, the style of writing is hard to be comprehended or followed.

This was starting from the Title until the end of Ms

For example: title: Equol effect on.... : this is better saif The effect of Equol, an isoflavone, on .....

This is since this flavone is not very common or known.

Kindly consult a colleague proficient in English and academic writing to help you better demonstrate your idea

We have submitted the revised manuscript to the academic editing service for language edit.

---

## [Decision Letter · Decision Letter 1]

31 Aug 2021

Effects of an equol-containing supplement on advanced glycation end products, visceral fat and climacteric symptoms in postmenopausal women: A randomized controlled trial

PONE-D-21-06650R1

Dear Dr. Myint,

We’re pleased to inform you that your manuscript has been judged scientifically suitable for publication and will be formally accepted for publication once it meets all outstanding technical requirements.

Kind regards,

Walid Kamal Abdelbasset, Ph.D.

Academic Editor

PLOS ONE

Additional Editor Comments (optional):

Reviewers' comments:

Reviewer's Responses to Questions

**Comments to the Author**

1. If the authors have adequately addressed your comments raised in a previous round of review and you feel that this manuscript is now acceptable for publication, you may indicate that here to bypass the “Comments to the Author” section, enter your conflict of interest statement in the “Confidential to Editor” section, and submit your "Accept" recommendation.

Reviewer #1: All comments have been addressed

Reviewer #2: All comments have been addressed

2. Is the manuscript technically sound, and do the data support the conclusions?

Reviewer #1: (No Response)

Reviewer #2: Yes

3. Has the statistical analysis been performed appropriately and rigorously? 

Reviewer #1: (No Response)

Reviewer #2: Yes

4. Have the authors made all data underlying the findings in their manuscript fully available?

Reviewer #1: (No Response)

Reviewer #2: Yes

5. Is the manuscript presented in an intelligible fashion and written in standard English?

Reviewer #1: (No Response)

Reviewer #2: Yes

6. Review Comments to the Author

Reviewer #1: (No Response)

Reviewer #2: Paper titled (Effects of an equol-containing supplement on advanced glycation end products, visceral fat and climacteric symptoms in postmenopausal women: A randomized controlled trial) by Yoshikata et al. was improved and I recoomedn acceptance of the current form.

7. PLOS authors have the option to publish the peer review history of their article (what does this mean?). If published, this will include your full peer review and any attached files.

Reviewer #1: No

Reviewer #2: **Yes: **Sawsan A. Zaitone

---

## [Editor Report · Acceptance letter]

2 Sep 2021

PONE-D-21-06650R1 

Effects of an equol-containing supplement on advanced glycation end products, visceral fat and climacteric symptoms in postmenopausal women: A randomized controlled trial 

Dear Dr. Myint:

I'm pleased to inform you that your manuscript has been deemed suitable for publication in PLOS ONE. Congratulations! Your manuscript is now with our production department. 

Kind regards, 

on behalf of

Dr. Walid Kamal Abdelbasset 

Academic Editor

PLOS ONE